# The Combined Effect of Hot Water Treatment and Chitosan Coating on Mango (*Mangifera indica* L. cv. Kent) Fruits to Control Postharvest Deterioration and Increase Fruit Quality

Hoda A. Khalil [1], Mohamed F. M. Abdelkader [2], A. A. Lo'ay [3,*], Diaa O. El-Ansary [4], Fatma K. M. Shaaban [5], Samah O. Osman [5], Ibrahim E. Shenawy [6], Hosam-Eldin Hussein Osman [7], Safaa A. Limam [8], Mohamed A. Abdein [9,*] and Zinab A. Abdelgawad [10]

1   Department of Pomology, Faculty of Agriculture (EL-Shatby), University of Alexandria, Alexandria P.O. Box 21545, Egypt; hoda.khalil@alexu.edu.eg
2   Department of Plant Production, College of Food and Agriculture, King Saud University, Riyadh 12372, Saudi Arabia; mohabdelkader@ksu.edu.sa
3   Pomology Department, Faculty of Agriculture, Mansoura University, Mansoura, El-Mansoura P.O. Box 35516, Egypt
4   Precision Agriculture Laboratory, Department of Pomology, Faculty of Agriculture (El-Shatby), University of Alexandria, Alexandria P.O. Box 21545, Egypt; diaa.elansary@alexu.edu.eg
5   Horticulture Research Institute, ARC, Giza 12619, Egypt; dr.fatmakorany@yahoo.com (F.K.M.S.); ayatosman012@gmail.com (S.O.O.)
6   Pomology Department, Faculty of Agriculture, Cairo University, Cairo P.O. Box 12613, Egypt; Dr.shenawy@hotmail.com
7   Anatomy Department, College of Medicine, Taif University, P.O. Box 11099, Taif 21944, Saudi Arabia; h.hussein@tu.edu.sa
8   Food Science and Technology Department, Faculty of Agriculture, Assiut University, Assiut 71515, Egypt; Limamsafaa@gmail.com
9   Biology Department, Faculty of Arts and Science, Northern Border University, Rafha 91911, Saudi Arabia
10  Botany Department, Women's College, Ain Shams University, Cairo 11566, Egypt; Zinababdelgawad@women.asu.edu.eg
*   Correspondence: Loay_Arafat@mans.edu.eg (A.A.L.); abdeingene@yahoo.com (M.A.A.)

**Abstract:** The synergistic effect of dipping in 55 °C for 5 min of hot water (HW) and 1% chitosan coating during the storage of mango at 13 ± 0.5 °C and 85%–90% relative humidity for 28 days was investigated. The combined treatment significantly suppressed the fruit decay percentage compared with both the single treatment and the control. In addition, the specific activities of key plant defense-related enzymes, including peroxidase (POD) and catalase (CAT), markedly increased. The increase occurred in the pulp of the fruits treated with the combined treatment compared to those treated with HW or chitosan alone. While the control fruits showed the lowest values, the combination of pre-storage HW treatment and chitosan coating maintained higher values of flesh hue angle (h°), vitamin C content, membrane stability index (MSI) percentage, as well as lower weight loss compared with the untreated mango fruits. The combined treatment and chitosan treatment alone delayed fruit ripening by keeping fruit firmness, lessening the continuous increase of total soluble solids (TSS), and slowing the decrease in titratable acidity (TA). The results showed that the combined application of HW treatment and chitosan coating can be used as an effective strategy to suppress postharvest decay and improve the quality of mango fruits.

**Keywords:** hot water; chitosan; mango; decay; storage; membrane stability index

## 1. Introduction

Mango fruits (*Mangifera indica* L.) are recognized as one of the most desirable fruits due to their appealing color, delectable flavor, and superior nutritional value. However, mangoes, a climacteric fruit, ripen shortly after harvest and are susceptible to anthracnose

caused by *Colletotrichum* species, which results in significant postharvest losses and restrictions on mango fruit storage, handling, and transportation [1]. Mango fruits are currently stored in a controlled (or modified) atmosphere to control postharvest decay and delay the ripening process [2]. In addition, fungicides are widely used to minimize postharvest decay and prolong the shelf life of mango fruits. However, fungicides are being limited as pathogens gain resistance to them, and consumers are concerned about the risks associated with fungicide residue [3]. Therefore, alternative and safe techniques are needed to slow the ripening of mango fruit and reduce postharvest decay.

Hot water (HW) treatment is the oldest and simplest form of heat treatment for controlling postharvest decay that uses a combination of appropriate temperatures (typically over 40 °C) and exposure durations to avoid fruit quality loss [4,5]. In a variety of fruits, HW treatment can effectively inhibit many important postharvest pathogens [6]. Treatment with HW at 55 °C for 35 min reduced the incidence of anthracnose in mango cultivars 'Tu Shien' [7], 'Kent', 'Keitt', and 'Tommy Atkins' [8]. Likewise, Dessalegn et al. [6], working on mango cv. 'Amba Kurfa', found that HW treatment at 51 °C for 3 min decreased the amount of anthracnose disease. Additionally, HW treatment has been identified as an elicitor for the activation of the defensive response in harvested fruits [9].

Although HW treatment helps prevent postharvest decay in mango fruits, there have been reports of detrimental impacts on the quality of the fruit, including accelerated fruit ripening, fruit skin browning, and mango fruit softening [10]. Based on the results presented above, it would be better to develop new treatments that may mitigate the negative effects of HW treatment, prevent postharvest decay, and delay the ripening process of mango fruits.

Several biopolymers, including chitosan, pectin, alginate, starch, carrageenan, zein, soy protein, and gelatin, have been applied in the development of coating formulations for fruit shelf life extension. Edible coating is simple, biodegradable, and ecologically friendly, it is a good alternative for synthetic materials, and it may be consumed by humans [11]. Edible coatings have been highlighted as a potential technology to prevent postharvest infection and the associated fungal degradation of fruits [12–14]. The application of fruit coatings has demonstrated technological advantages such as better appearance, antibacterial and antioxidant properties, and improved taste [15]. Some coatings have already been tested on tropical fruits, including avocados and mangoes, with different degrees of effectiveness. For instance, Daisy et al. [16] found that gum Arabic (15%) preserved ascorbic acid and carotenoids in 'Apple' mango kept at room temperature for 15 days. Likewise, Moalemiyan et al. [17] reported that coating mango with pectin, sorbitol, monoglyceride, and beeswax combinations resulted in an increased shelf life, especially at decreased color development, weight loss, softness, and acid production compared with the control. In addition, a study by Bambalele et al. [15] reported that moringa leaf extract (1%) and carboxymethyl cellulose (1%) maintained the ascorbic acid and membrane integrity and delayed fruit softening in 'Keitt' mango after storage at 10 °C for 21 days.

Chitosan is one of the polysaccharide-based coatings. It is a high molecular weight cationic polysaccharide commonly formed by the alkaline deacetylation of chitin found in the crustacean exoskeleton, fungal cell walls, and other biological components [18]. It is composed of poly-1,4-β-D-glucopyranos amine and 2-amino-2-deoxy-(1->4)-β-D-glucopyranan. Chitosan has great potential as a film or a biodegradable edible coating for food packaging [19], with good biocompatibility, nontoxicity [18], and film-forming characteristics [20]. Chitosan has been used on a variety of fruits such as mango as a semipermeable coating to prolong storage life and decrease postharvest decay [21,22]. The application of chitosan in mango [23] has been demonstrated to enhance fruit quality, keep firmness, decrease ethylene production and mold contamination, delay the ripening process and senescence, and decrease color changes.

The combination of edible coating and HW treatment has been examined to maintain fruit quality and minimize unanticipated damage [24,25]. Keeping fruit quality and controlling postharvest decay in fruit cannot be entirely controlled by HW treatment or

chitosan treatment alone. For several reasons, combining HW and chitosan may have a synergistic impact on fruit: (1) The fruit surface may be partly disinfected by the HW treatment; (2) Pathogen resistance may be induced by chitosan; (3) Combining HW with chitosan can improve the effectiveness of postharvest disease control. In the previous studies, researchers used a combination of these two treatments on sweet cherry [26], papaya [27], and dragon fruit [28], and the results indicated that the combination treatment reduced postharvest disease and preserved fresh fruit quality better than HW or chitosan.

Although HW and edible coatings have been intensively investigated in recent years, the combined effect of these treatments has received less attention, especially on mango fruits. To our knowledge, there are no published data about the use of hot water treatment and chitosan coatings for maintaining fruit quality and prolonging the shelf life of "Kent" mangoes. Thus, our research aimed to see if HW treatment, followed by chitosan coating, may help keep mango fruit fresh, maintain quality indices, and extend the shelf life of "Kent" mangoes while also reducing postharvest decay. In addition, HW treatment-induced resistance in mango fruit was also investigated to better understand the defense mechanism of HW treatment against pathogens during storage.

## 2. Materials and Methods

### 2.1. Plant Materials and Treatments

The present study was carried out during the 2020 and 2021 seasons on mature green stage mango fruits (*Mangifera indica* L. cv. "Kent"). The fruits were harvested from a private orchard at Alexandria–Cairo desert road and were immediately transported to the Horticulture Lab in the Faculty of Agriculture, Alexandria University, Egypt. The coating solution preparation was done as 1 g of chitosan (40 kDa in thickness, from crab shells) which was gently dissolved in 100 mL of 2 percent acetic acid solution (*v/v*) on a magnetic stirrer (350 rpm INTLLAB, New York, NY, USA) to get 1 percent chitosan solution (*w/v*), and Tween 20 (0.2 g) was added in the chitosan solution. The fruits were selected for uniformity of size, ripeness, and being free of defects, and were divided into four groups (50 fruit for each). The first group of fruits was washed with distilled water (control). The second group was dipped in hot water (HW) at 55 °C for 5 min. The third group was dipped in 1% chitosan solution for 1 min. The fourth group was dipped in hot water (HW) at 55 °C for 5 min, and then dipped in 1% chitosan solution for 1 min. After being air dried, all the fruits were stored at 13 °C, 85%–90% RH for 28 days. The initial physio-chemical properties were determined in ten mango fruits and the changes were followed up in 7 day intervals during the storage period.

### 2.2. Measurements of Fruit Physical and Chemical Features

#### 2.2.1. Decayed Fruit Percentage

Decay due to browning skin, shriveling, and diseases were recorded and estimated based on the initial number of fruits in each sample and reported as a percentage.

#### 2.2.2. Fruit Firmness

Fruit firmness was measured using an Effegi pressure tester (Effegi, 48011 Alfonsine, Alfonsine, Italy) connected to a flat probe (8 mm diameter). At the equator of the fruit, the skins were removed in four pieces and four independent measurements were recorded for each fruit.

#### 2.2.3. Weight Loss Percentage

Weight loss was assessed by weighing ten labeled treated fruits at 7-day intervals during the storage. The percentage of weight loss was assessed by the following equation:

$$\text{Weight loss \%} = \frac{\textit{The initial weight} - \text{fruit weight at examination date}}{\textit{The initial weight}} \times 100$$

### 2.2.4. Fruit Content of Total Soluble Solids (TSS) and Titratable Acidity (TA)

For each treatment, four fruit pulp samples were squeezed out, and the resulting juice was used to measure the TSS percentage using a hand refractometer (Atago, Japan), and the titratable acidity in grams of citric acid per 100 mL of fruit juice [29].

### 2.2.5. Vitamin C

Vitamin C was determined by oxidizing ascorbic acid with 2,6-dichlorophenol en-dophenoldye and the results were reported in mg/100 g on a fresh weight basis [30].

### 2.2.6. Fruit Color Index

Flesh color was measured using a Minolta Chroma Meter CR-200 (Minolta Co. Ltd., Osaka, Japan). Flesh color measurements were expressed as hue angle chromaticity values (h°). Four readings were obtained at different points on each mango fruit for each data observation [31,32].

### 2.2.7. Defense-Related Enzymes Activities

The peroxidase activity of *M. indica* kernels was determined using a colorimetric test based on the initial increase in absorbance at 420 nm in the presence of a constant volume of hydrogen peroxide and a crude extract of pyrogallol. POD activity was expressed as U/g FW, where one unit of peroxidase activity was defined as the amount of extract that caused a 0.001 per minute change in absorbance at 420 nm [33].

Catalase (CAT) activity was measured using the method of Beers and Sizer [34], with certain modifications. The reaction mixture consisted of 0.2 crude extracted from 50 mM sodium phosphate buffer (pH 7.0) and 150 mL of 20 mM of $H_2O_2$. The action of CAT on hydrogen peroxide caused a decrease in absorbance at 240 nm. The change in absorbance per minute was defined as one unit of CAT.

### 2.2.8. Membrane Stability Index Percentage (MSI)

Ion leakage from fruit peels was assessed in peel discs using the method described by Sairam et al. [35], with modification, and was represented as a percentage of membrane stability index (MSI). For each replicate/treatment, 3 g of wash disks were randomly selected and placed in 30 mL of deionized water at room temperature on a shaker for 4 h. Before boiling ($C_1$), a digital conductivity meter (Orion 150A +, Thermo Electron Corporation, Colorado, CO, USA) was used to check the conductivity. The same disk was placed in a boiling water bath (100 °C) for 30 min to release all electrolytes, cooled to 22 ± 2 °C in running water, and boiled to measure conductivity ($C_2$). MSI was calculated as a percentage using the following formula: $[1 - (C_1/C_2)] \times 100$. Fruit softening or the appearance of chilling injury signs indicated the termination of the trial.

### 2.3. Statistical Analysis

The two-way analysis of variance (ANOVA) method was used to evaluate the data for the effects of the treatments on the investigated parameters. According to Snedecor and Cochran [36], the treatment means were separated and compared using the least significant differences (L.S.D.) at the 0.05 level of significance. SPSS 18.0 software was used for all statistical analyses (SPSS Inc., Chicago, IL, USA). In the figures, data were presented as means of standard errors (SE).

## 3. Results

### 3.1. Effect of Chitosan and Hot Water (HW) Treatments on Fruit Quality

#### 3.1.1. Decay

As for the storage period, regardless of the treatments, the obtained results showed that the incidence of fruit decay appeared after 14 days of cold storage, followed by a consistent increase with increasing the storage period up to 28 days (Tables 1 and 2, Figure 1). Data also indicated that all treatments significantly decreased the decay incidence of mango fruits

during the two seasons of the study. The chitosan treatment did not significantly differ from the combined treatment in both seasons. The incidence of decay was higher in the control mango fruits than in the other treatments. The control fruit had an 8.9% decay incidence after 14 days of storage, but the HW treatment, chitosan, and the combination treatments had no decay incidence (Table 1). In the first season, at the end of the storage date, the control treatment had a decay incidence of 30.20%, followed by the HW-treated samples (19.35%), while the chitosan and combined treatments were only 3.6 and 2.4%, respectively (Table 1). A significant interaction effect between the treatments and the storage period on the percentage of fruit decay was obtained in both seasons. Data revealed that the fruit decay percentage was least with all treatments, especially when those treatments were accompanied by the shortest storage period in comparison with the same treatments accompanied by the longest storage period.

**Table 1.** Effect of hot water treatment and chitosan coating on postharvest decay (%) of mango fruits during storage at 13 ± 0.5 °C and 85%–90% relative humidity (RH) for 28 days during 2020 season.

| Treatments | Storage Period (Days) | | | | | |
|---|---|---|---|---|---|---|
| | **0** | **7** | **14** | **21** | **28** | **Mean** |
| | Season 2020 | | | | | |
| **Control** | **0.00 h** | **0.00 h** | **8.96 d** | **17.5 c** | 30.20 a | 11.33 a |
| Hot water (HW) | 0.00 h | 0.00 h | 0.00 h | 6.13 e | 19.35 b | 5.09 b |
| 1% chitosan | 0.00 h | 0.00 h | 0.00 h | 0.00 h | 3.68 f | 0.73 c |
| HW + 1% chitosan | 0.00 h | 0.00 h | 0.00 h | 0.00 h | 2.43 g | 0.48 c |
| Mean | 0.00 d | 0.00 d | 2.24 c | 5.90 b | 13.91 a | – |

Means followed by the same letters within treatments, storage period, and their interactions in 2020 season are not significantly different at level $p \leq 0.05$.

**Table 2.** Effect of hot water treatment and chitosan coating on postharvest decay (%) of mango fruits during storage at 13 ± 0.5 °C and 85%–90% relative humidity (RH) for 28 days during 2021 season.

| Treatments | Storage Period (Days) | | | | | |
|---|---|---|---|---|---|---|
| | **0** | **7** | **14** | **21** | **28** | **Mean** |
| | Season 2021 | | | | | |
| **Control** | **0.00 h** | **0.00 h** | **7.58 d** | **15.66 c** | 28.32 a | 10.31 a |
| Hot water (HW) | 0.00 h | 0.00 h | 0.00 h | 6.30 e | 21.50 b | 5.56 b |
| 1% chitosan | 0.00 h | 0.00 h | 0.00 h | 0.00 h | 4.13 g | 0.82 c |
| HW + 1% chitosan | 0.00 h | 0.00 h | 0.00 h | 0.00 h | 5.12 f | 1.02 c |
| Mean | 0.00 d | 0.00 d | 1.89 c | 5.49 b | 14.76 a | – |

Means followed by the same letters within treatments, storage period, and their interactions in 2021 season are not significantly different at level $p \leq 0.05$.

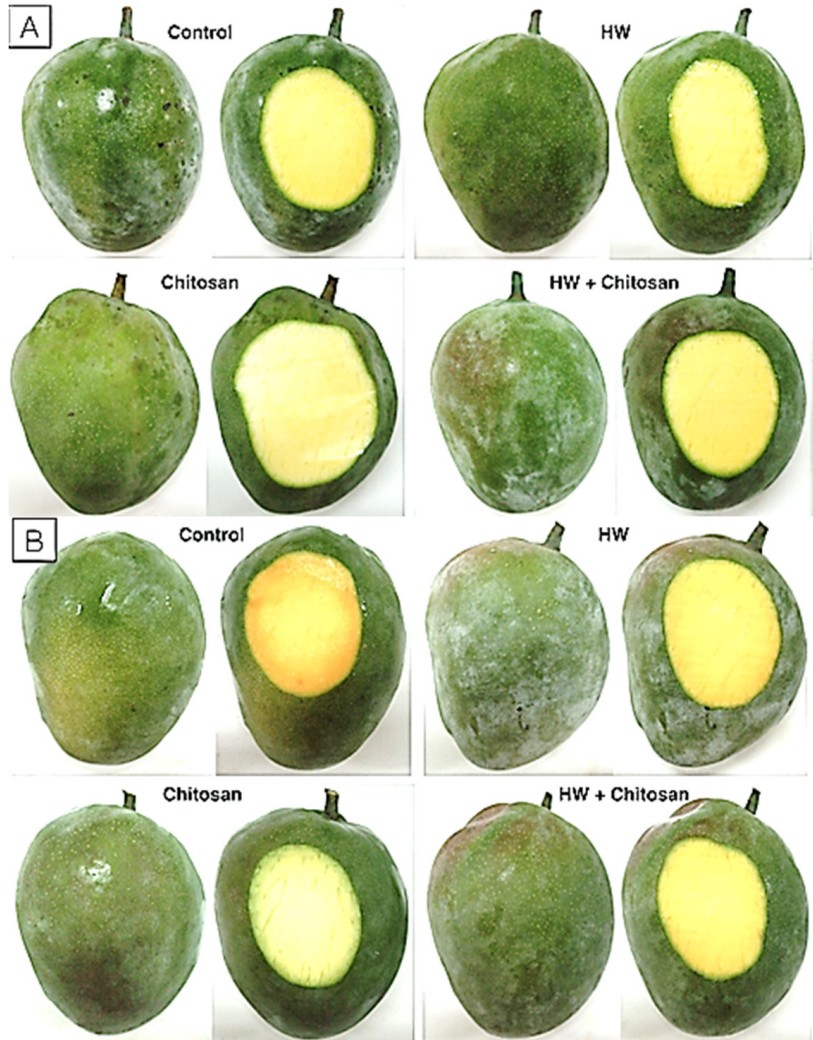

**Figure 1.** Visual appearance of mango fruit after 7 days (**A**) and 21 days (**B**) of storage at $13 \pm 0.5\ °C$ and 85%–90% relative humidity.

3.1.2. Weight Loss

The percentage of mango fruit weight loss increased dramatically when the storage duration was extended to 28 days (Figure 2A and Table 3). All treatments considerably decreased weight loss of 'Kent' mango fruits during cold storage when compared to untreated control fruits (Figure 2A and Table 3). Fruit treated with chitosan or the combined treatment (Figure 2A) lost less weight (4.13% and 5.16%, respectively) than the control or the HW-treated fruit (6.80% and 6.16%, respectively). As for the interaction effect, data showed that all the treatments at the same storage period decreased weight loss.

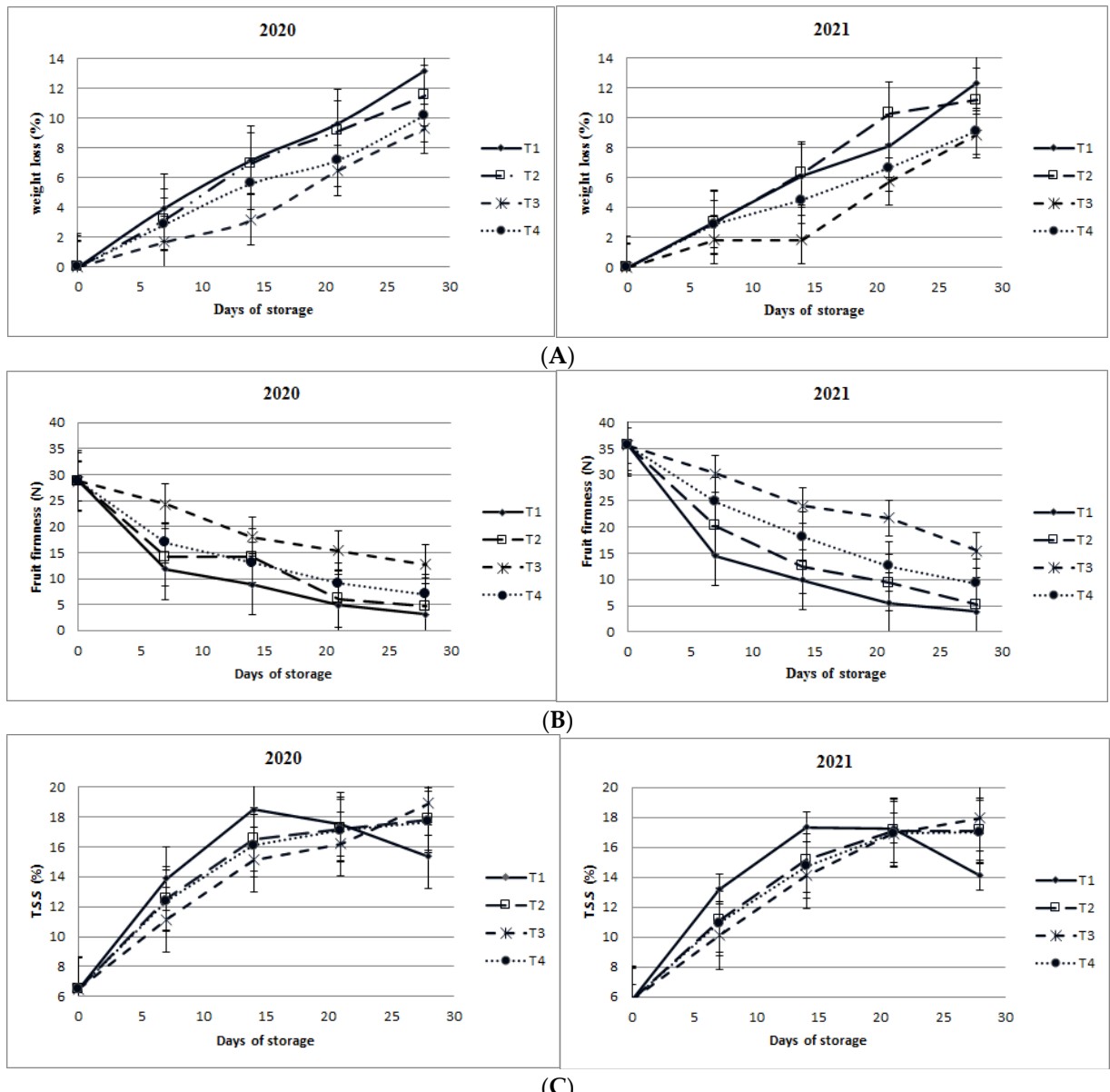

**Figure 2.** Effect of hot water treatment and chitosan coating on weight loss (**A**), fruit firmness (**B**), and total soluble solids (T.S.S) (**C**) of mango fruits during storage at 13 ± 0.5 °C and 85%–90% relative humidity (RH) for 28 days during 2020 and 2021 seasons. Values are means ± SE from three replicates. Statistical analysis was performed using LSD test.

**Table 3.** Results of the analysis of variance with mean square testing the effects of treatments (T), storage period (S), and their interactions on fruit firmness (N), weight loss (%), TSS (%), acidity (%), vitamin C (mg/100 g FW), color index (h°), peroxidase activity (units/mg FW), catalase activity (units/mg FW), and MSI (%) during 2020 and 2021 seasons.

| Season 2020 | Fruit Firmness (N) | Weight Loss (%) | TSS (%) | Acidity (%) | Ascorbic Acid (mg/100 g FW) | Color Index (h°) | Peroxidase Activity(Units/mg FW) | Catalase Activity (Units/mg FW) | MSI (%) |
|---|---|---|---|---|---|---|---|---|---|
| Treatments (T) | 38.00 *** | 38.20 *** | 3.17 * | 969.00 *** | 21 *** | 121.40 *** | 199.80 *** | 3.39 * | 199.80 *** |
| Storage period (S) | 414.30 *** | 416.30 *** | 505 *** | 8362.10 *** | 541.50 *** | 1755.10 *** | 771.80 *** | 73.11 *** | 771.80 *** |
| T X S | 3.74 *** | 3.70 *** | 7.77 *** | 118.80 *** | 2.32 * | 13.70 *** | 58.80 *** | 7.80 *** | 58.80 *** |

| Season 2021 | Fruit Firmness (N) | Weight Loss (%) | TSS (%) | Acidity (%) | Ascorbic Acid (mg/100 g FW) | Color Index (h°) | Peroxidase Activity (O.D) | Catalase Activity | MSI (%) |
|---|---|---|---|---|---|---|---|---|---|
| Treatments (T) | 158.40 *** | 4.25 * | 2.77 ns | 389.20 *** | 191.30 *** | 332.80 *** | 52.70 *** | 25.26 *** | 52.70 *** |
| Storage period (S) | 600.10 *** | 90.87 *** | 701.28 *** | 2349.80 *** | 3115.30 *** | 3395.80 *** | 67.30 *** | 37.51 *** | 67.30 *** |
| T X S | 12.30 *** | 2.89 ** | 13.19 *** | 61.90 *** | 18.80 *** | 32.58 *** | 13.80 *** | 2.90 ** | 13.80 *** |

ns, *, **, *** nonsignificant, or significant at $p$ = 0.05, 0.01, and 0.001, respectively.

### 3.1.3. Firmness

Data presented in Table 3 and Figure 2B reflected the reduction in fruit firmness with the progress of the storage period for all treatments. However, after 7 days of storage, the control fruits and the fruits treated with HW softened faster than the other treatments. At the end of storage, the control fruit and the fruit treated with HW showed low values of 11.48 and 12.53 N, respectively, whereas the combined-treated fruits and the chitosan-treated fruits showed a high fruit firmness value of 14. 96 and 19.84 N, respectively. Data also showed that there was a significant interaction between the treatments and the storage period for mango fruit firmness in both seasons. The ripening of mango fruits leads to a loss in firmness with the progress of the storage period.

### 3.1.4. Total Soluble Solids (TSS) and Titratable Acidity (TA)

Total soluble solids and titratable acidity, as well as a comparison of the means over the storage period for both experimental seasons, are shown in Figures 2C and 3A and Table 3. TSS increase in all treatments over time, while TA decreased over the fruit's storage period, which is a climacteric fruit feature during the ripening process. At the end of the storage period, all postharvest treatments significantly increased TSS and decreased TA compared to the control treatment. In both seasons, fruits treated with chitosan had the lowest TSS values (13.57 and 12.97, respectively) and the highest TA values (0.79 and 0.77, respectively).

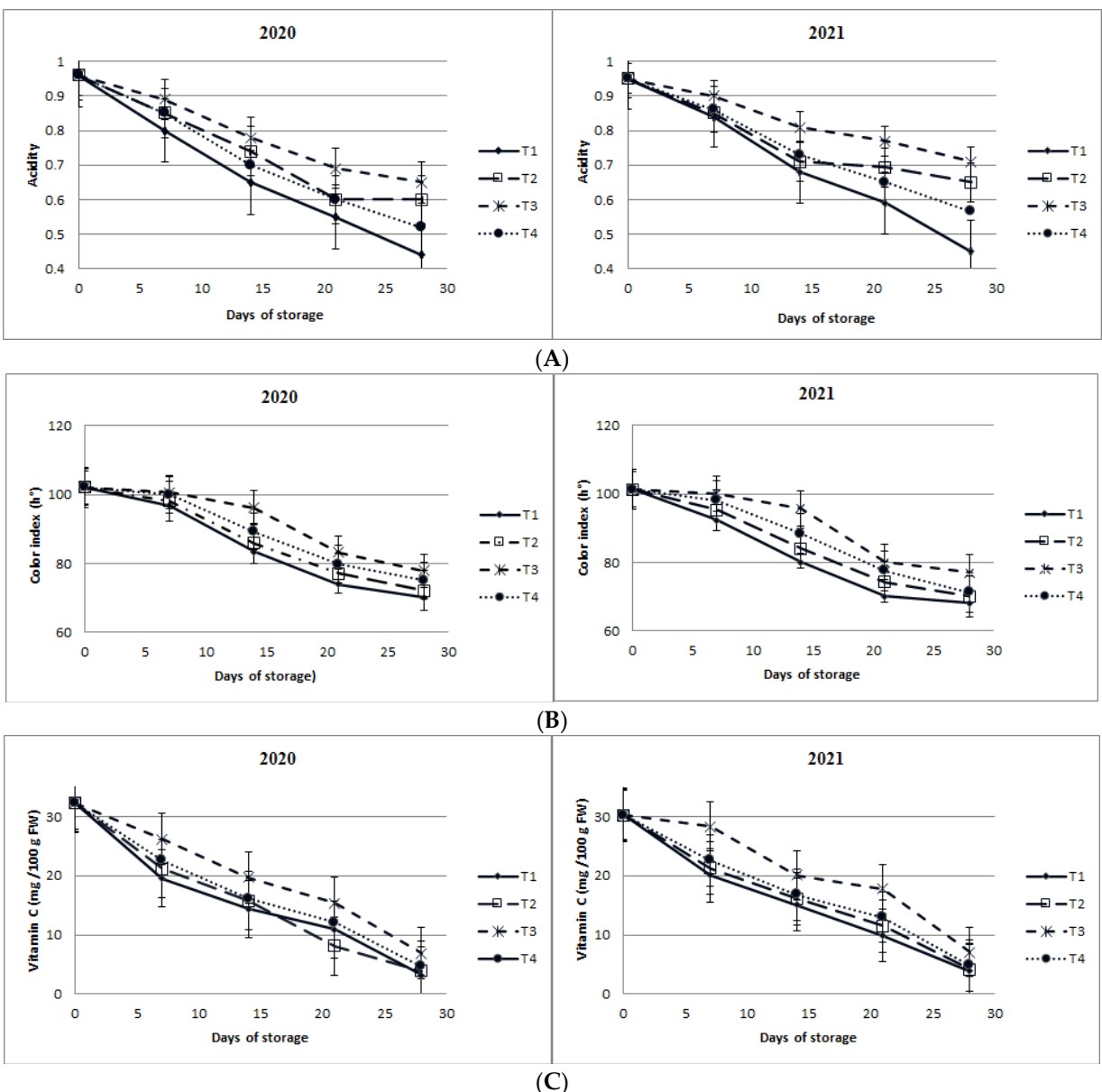

**Figure 3.** Effect of hot water treatment and chitosan coating on acidity (**A**), color index (**B**), and vitamin C (**C**) of mango fruits during storage at 13 ± 0.5 °C and 85%–90% relative humidity (RH) for 28 days during 2020 and 2021 seasons. Values are means ± SE from three replicates. Statistical analysis was performed using LSD test.

### 3.1.5. Changes in Flesh Color (h°)

Data shown in Figures 1 and 3B and Table 3 presented the changes in flesh hue angle according to chitosan, HW treatment, and the cold storage period. Chitosan-treated fruits had the highest significant values of flesh hue angle, followed by the combined treatment and HW, whereas the control treatment showed a lower flesh hue angle content than the other treatments.

### 3.1.6. Changes in Vitamin C Content, Peroxidase (POD) Activity, Catalase (CAT) Activity, and Membrane Stability Index Percentage (MSI%)

Peroxidase and catalase activities increased during the storage period in all treatments. However, after one week of the storage period, POD and CAT activities were lower in all treatments, including the control (Figures 3C and 4A–C, and Table 3). At the end of the storage period, all the treatments showed higher POD and CAT activities than the control.

The chitosan and combined treatment presented the highest POD and CAT activities. Vitamin C content was lower than initial in all treatments and decreased during storage. At the end of the storage period (28 days), all treatments obtained a higher vitamin C content than the control. However, the chitosan and combined treatments showed higher vitamin C content than the other treatments. In all treatments, the membrane stability index (MSI) recorded lower values than initially and declined throughout the storage period (Figure 4C and Table 3). All treatments applied at the end of the storage period maintained a higher MSI than the control. The chitosan treatment and the combination treatment had a greater MSI in both seasons than the other treatments, including the control.

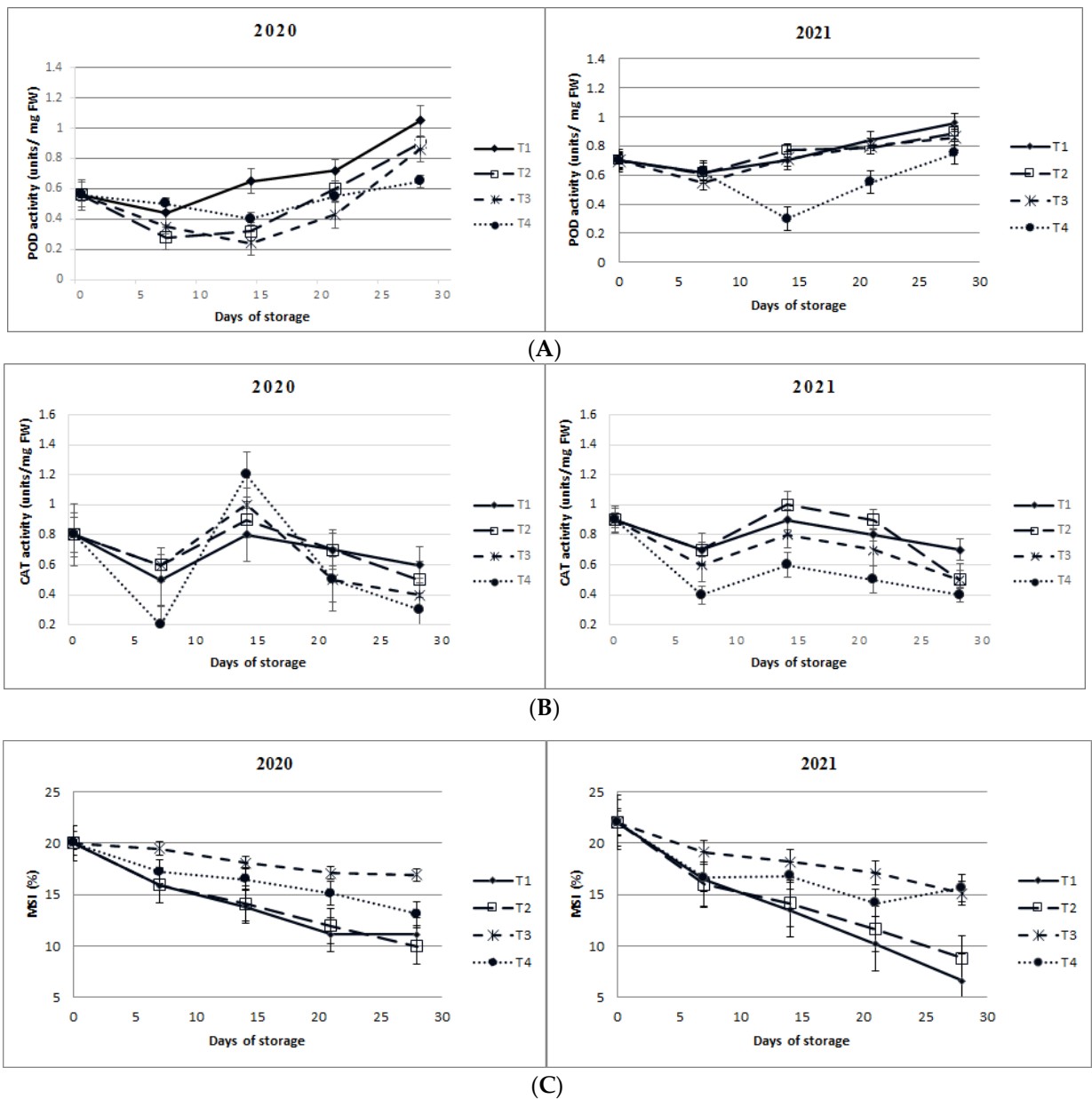

**Figure 4.** Effect of hot water treatment and chitosan coating on peroxidase (POD) activity (**A**), catalase (CAT) activity (**B**), and membrane stability index (MSI) percentage (**C**) of mango fruits during storage at $13 \pm 0.5\,^\circ$C and 85%–90% relative humidity (RH) for 28 days during 2020 and 2021 seasons. Values are means $\pm$ SE from three replicates. Statistical analysis was performed using LSD test.

## 4. Discussion

As a climacteric fruit, mango fruit exhibits relatively high levels of bioactivity, including high respiration rates and postharvest ethylene production, thus reducing its shelf life. Due to consumer concerns about the use of synthetic chemicals, various natural coatings are currently being investigated throughout the shelf life for their effectiveness in slowing ripening and maintaining fruit quality [37–39]. Our investigation demonstrated that chitosan coating, alone or in combination with HW treatment, could efficiently delay ripening, increase postharvest quality, and control the decay of mango fruits. These results revealed that chitosan coating could be an alternate and effective technique for prolonging the postharvest life of mango fruits. In the current study, chitosan coating, alone or combined with HW treatment, significantly decreased the decay percentage after 3 weeks of cold storage compared to the control and the other treatments (Tables 1 and 2). The fruit coating with a concentration of 0.2% chitosan significantly inhibited the decay incidence of mango fruits caused by disease and maintained the fruit quality [5]. Chitosan postharvest application is known to influence the host–pathogen as antibacterial and antifungal activity [22,40]. It can destroy the plasma membrane of the spore of several pathogens, inhibit mycelial growth, and induce damage to the fungal cytoplasm [41]. Moreover, chitosan may also induce a defense mechanism in host tissues [42]. Several studies have shown that the use of HW is beneficial for tropical fruits [43–46]. The main cause of fruit weight loss is water loss induced by respiration and transpiration processes [47]. Chitosan has been reported to decrease the respiration rate in mango fruit [47–49]. The lower weight loss observed in chitosan-treated fruits may be related to the higher vapor barrier of chitosan. On the contrary, one study reported that 'Ataulfo' mango coated with chitosan film had greater weight loss [50]. Chitosan coating is reported to minimize transpiration losses by forming a semipermeable layer on the fruit's surface and acting as a selective permeability to water vapor [51]. In addition, it is reported to decrease the transpiration rate and retard senescence by modifying the internal atmosphere of the fruits [52]. The application of HW treatment has been reported to reduce or increase the weight loss of fruits. In this study, the HW treatment decreased fruit weight loss. Consistent with our result, Fawaz [5,53] mentioned that the weight loss of 'Alphones' mango fruits was reduced by a 45 °C HW treatment, suggesting that a mild heat treatment would dissolve the cuticle wax and decrease water loss [54]. The ripening of mango fruit is marked by a softening of the texture and a change in the color of the surface. The results of our study showed that the chitosan coating alone or in combination with the HW treatment effectively slowed the ripening of mango fruit, as evidenced by the retention of firmness and the delayed color change. In addition, TA and vitamin C reduction, TSS increase, and the weight loss of mango fruit were significantly suppressed by the chitosan coating. Thus, the application of the chitosan coating alone or in combination with the HW treatment effectively maintained the postharvest quality of the mango fruits according to the result obtained on the mango fruits treated by chitosan-based coating after harvest [49,55]. In the previous studies, chitosan maintained the firmness of the mango fruit [47,56,57]. Amin et al. [58] reported that mango fruit firmness was found to decrease linearly with storage time. The rate of firmness loss, however, was consistently decreased with the addition of a up to 2% chitosan–Aloe vera coating. The results of the HW treatment are consistent with [9,43]. The slowing of the softening might be attributed to the prevention of the formation of cell wall hydrolysis enzymes, which maintain membrane stability and reduce firmness [59]. Consistent with our result, chitosan has been shown to reduce fruit TSS during storage in mango [47,56,60]. Chitosan-treated fruits had the highest significant values of flesh hue angle, followed by the combined treatment and the HW treatment. These findings are consistent with those of Zhu et al. and Djioua et al. [25,47] on mango. They reported that the application of a 2% chitosan coating and a HW treatment at 50 °C for 30 min were effective treatments to maintain firmness and delay color change during fruit storage. Chitosan formed a semipermeable layer over the fruit peel and altered the atmosphere by elevating $CO_2$ and lowering $O_2$, which inhibited ethylene production and delayed ripening [61]. As the ripening process was delayed, it consequently reduced

the color changes by decreasing carotenoid biosynthesis and preserving the chlorophyll content. The hue values decreased with the advancing of the storage period and the values were significant for all treatments, with the change of flesh color from creamy to orange. Previously, the use of 1% chitosan was shown to improve the ascorbic acid content of mango [57,62]. Similar results have been reported in a previous study using the HW treatment [5]. The ascorbic acid content was found to be increased 1.14-fold in HW-treated fruits in contrast with the control fruit. The results showed that the chitosan treatment and the combination treatment had a greater MSI in both seasons than the other treatments, including the control (Figure 3C and Table 3). Ripening is a potentially oxidative process in which the transition from the maturing stage to the ripening stage is accompanied by a dynamic move to an oxidative state [63]. Likewise, excessive ROS generation can lead to the oxidation of the cell membrane lipids and proteins involved in mango ripening, which results in a gradual loss of membrane stability because of the changes in the biophysical and biochemical characteristics of the cell membranes. The expression of genes encoding enzymes involved in the fruit antioxidant system, including POD, PPO, and catalase, increases during ripening [64,65], and endogenous defense against the accumulation of harmful ROS has also been reported [49,66,67]. This might demonstrate that these treatments, especially chitosan and combined treatments, improved the antioxidant network of the fruit [67,68], allowing for the more effective regulation of metabolic free radical levels, hence preserving peel cell membrane integrity and maintaining better flesh firmness. Although many of the previous publications have indicated that both hot water and chitosan could be applied for different fruit protections, the novelty of our study is that, under the synergistic effect of the coating of the chitosan solution with a low concentration (1%) and the hot water treatment, the postharvest decay was suppressed, the quality of the mango fruits was improved, and the shelf life was extended by increasing the temperature of the hot water and shortening the treatment time of the hot water. Moreover, different fruits have different profiles regarding their storage capacity, with different treatment applications.

## 5. Conclusions

Edible coatings combined with HW treatment were used to induce fruit decay resistance and improve fruit quality parameters. The present study evaluated the effect of a chitosan edible coating combined with a HW treatment on mango fruits during the storage time for 18 days at $13 \pm 0.5$ °C and 85%–90% relative humidity for 28 days. The results revealed that a combination of HW and chitosan treatments dramatically decreased the decay incidence percentage and improved the quality in mango fruits while also elevating the specific activity of POD and CAT defense-related enzymes. The combination of the prestorage HW treatment and chitosan coating maintained higher values of flesh hue angle (h°), vitamin C content, membrane stability index (MSI) percentage, as well as lower weight loss compared with the untreated mango fruits. Fruits treated with HW ripened the fastest, comparable to the control fruits and the other treatments. However, a combination of the HW and chitosan treatments slowed down the fruits' ripening. As a result, combining HW and the chitosan coating improved the effects of each treatment alone. This application might be a promising technology and a novel strategy for controlling fruit decay, thus maintaining mango fruit quality during the storage period.

**Author Contributions:** Conceptualization, M.F.M.A., S.O.O., I.E.S., H.-E.H.O. and Z.A.A.; Data curation, H.A.K., D.O.E.-A., F.K.M.S., S.O.O., I.E.S., S.A.L., M.A.A. and Z.A.A.; Formal analysis, S.A.L., M.A.A. and Z.A.A.; Funding acquisition, I.E.S. and H.-E.H.O.; Investigation, H.A.K.; Methodology, H.A.K., M.F.M.A., D.O.E.-A., F.K.M.S., S.O.O. and Z.A.A.; Project administration, D.O.E.-A. and F.K.M.S.; Resources, M.F.M.A., S.A.L. and Z.A.A.; Software, A.A.L., F.K.M.S., S.O.O., M.A.A. and Z.A.A.; Supervision, A.A.L., H.-E.H.O. and M.A.A.; Visualization, H.-E.H.O. and M.A.A.; Writing—original draft, M.F.M.A. and D.O.E.-A. All authors have read and agreed to the published version of the manuscript.

**Funding:** The authors extend their appreciation to Taif University for supporting this work, the Researchers Supporting Project under Project No. (TURSP-2020/116), Taif University, Taif, Saudi Arabia.

**Institutional Review Board Statement:** Not applicable.

**Informed Consent Statement:** Not applicable.

**Data Availability Statement:** Relevant data applicable to this research are within the paper.

**Acknowledgments:** The authors extend their appreciation to Taif University for supporting current work by Taif University Researchers Supporting Project number (TURSP-2020/116), Taif University, Taif, Saudi Arabia.

**Conflicts of Interest:** The authors declare no conflict of interest.

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
