# Peer review of "The Combined Effect of Hot Water Treatment and Chitosan Coating on Mango (Mangifera indica L. cv. Kent) Fruits to Control Postharvest Deterioration and Increase Fruit Quality"

_coatings, doi:10.3390/coatings12010083_

Round 1

Reviewer 1 Report

There are a series of problems in mango storage, such as black rot and self acceleration. As a bio polymer, chitosan, whose coating has certain advantages, such as safety. The authors’ research has a certain possibility of use, but the corresponding cost problem is one of the barriers for future application. In addition, the sample size discussed by the author may be statistically insignificant, and more samples are supposed in the future. More importantly, the authors’ description is quite subjective, such as "dramatically", hoping that the authors can describe the facts more objectively. It is suggested that the manuscript may be revised before reconsideration, and the revision contents include:

  1. It is necessary to provide sample photos, especially the progress of the control group and the experimental group during the observation period;
  2. Thickness and composition analysis of chitosan coating;
  3. Objectification of statement description
  4. Cost analysis, practicability and future prospects.

Author Response

Reviewer # 1

  1. We provided sample photos for the progress of the control group and the experimental group during the observation period. Please see Figure 1.
  2. chitosan (40 kDa thickness, from crab shells, it is composed of poly-1.4- β -D-glucopyranosamine; 2-Amino-2-deoxy-(1->4)- β -D-glucopyranan).
  1. Objectification of statement description…… Our results were clear, and we presented in our figures
  2. Cost analysis, practicability, and prospects……The coating treatments was in low cost compared to the fruit loss cost and our treatments minimized the fruit loss during the experiment

Reviewer 2 Report

The manuscript reports an investigation on the synergistic effect of hot water and chitosan coating on induction of fruit decay resistance and improvement of quality parameters of harvested mangoes. The paper needs some proofreadings and revisions. It requires major revisions before being considered for publication. Below, I include some specific comments.

  1. English language revision should be carried out.
  2. In the introduction, the novelty of the work should be better strengthened. What about similar coating? Other polymers such as pectin have already been investigated in order to extend the fruits storage.
  3. Figures need to be modified to make it better visible. The same font for axis labels must be used. Data are not sufficiently visible.
  4. Footnotes concerning the statistical analysis under eachtables should be added to make clearer the data comparison.
  5. What about the possibility of increasing the chitosan concentration?

Author Response

Reviewer # 2

  1. We revised English Language
  2. We strengthened the novelty of the work. And we provided literature about the similar coatings and its effect in extending fruit shelf life. Please check introduction section lines 81-98.
  3. We modified the Figures to make it better visible and we used the same font for axis labels.
  4. We added footnotes concerning the statistical analysis under the tables.
  5. About the possibility of increasing the chitosan concentration, From the previous literature, It is possible to increasing concentration more than in our study (1%) because chitosan is possibly safe. chitosan is a sugar that is obtained from the hard-outer skeleton of shellfish, including crab, lobster, and shrimp. It is used for medicine.

Reviewer 3 Report

Ref. No.: COATINGS-1520382

Title: The use of hot water treatment and chitosan coating on mango (Mangifera Indica L. cv. Kent) fruits to control postharvest deterioration and increase fruit quality

Hoda A. Khalil, Diaa O. El-Ansary Lo’ay, A. A, Fatma K. M. Shaaban, Samah O. Osman, Ibrahim E. Shenawy, Hosam-Eldin Hussein Osman, Mohamed A. Abdein and Zinab A. Abdelgawad

Attempt to prolong and ensure quality and microbial safety of mango (Mangifera Indica L. cv. Kent) fruits, to control postharvest deterioration and increase fruit quality, authors applied hot water treatment and chitosan coating. Results indicate that combined treatment gave the best results and possess potential to be applied for mango fruit protection.

Still, there are plenty of papers indicated that both hot water and chitosan, in synergy, could be applied for different fruit protection. Idea and methodology of all those papers are similar. So, I don’t find novelty of this paper.

I found the paper to be overall well written and much of it to be well described.

Title reflects the content of paper.

Authors used adequate literature for Introduction but in the part Reference there are several technical mistakes.

The experiment is clearly set up, and the analysis methods, their order, as well as description are mainly understandable.

In part Results, there are significant misleading in the marks of season years. It could be technical mistake or bad explanation in Materials and Methods.

Major comments:

Technical mistakes:

  • Hoda A. Khalil, Diaa O. El-Ansary Lo’ay, A. A, Fatma K. M. Shaaban, Samah O. Osman; Ibrahim E. Shenawy, Hosam-Eldin Hussein Osman, Mohamed A. Abdein and Zinab A. Abdelgawad
  • line 84: extra point after word „components“
  • line 85: extra space before [25]
  • Tables 1-4 referred seasons 2017 and 2018, while title of Tables, Figures 1-3, and Martials and method too, referees season 2020 and 2021. Are these mistakes or in Material and methods miss seasons 2017 and 2018?
  • References No:3, 7, 11,14, 30, 31, 35, 43,46, 50,58,61,65,70-72, 74 an d79 have technical miskateks

Also:

-line 117:  missing description of 1% chitosan preparation, or reference

- it is not clear why authors calculate means of different treatment and means of storage period

Author Response

Reviewer # 3

  1. Line 102: We removed extra point after word „components “.
  2. Line 105: We deleted extra space before the reference.
  3. Tables 1-4, We checked and corrected the mistake, the experiment was carried out during two consecutive seasons 2020 and 2021.
  4. We checked and corrected all references technical mistakes.
  5. Lines 135-139: We provided a chitosan preparation solution.
  6. We calculated the means of different treatments and means of storage period to determine the main effect of each factor separately on the decay % parameter.

Reviewer 4 Report

Article

The use of hot water treatment and chitosan coating on mango 2 (Mangifera Indica L. cv. Kent) fruits to control postharvest dete-3 rioration and increase fruit quality

Journal: Coating (MDPI)

Comments:

Line 2: modify title by clearing combine treatment or both single and combined

Line 24: concise the abstract

Line 28: complete sentence in one and half line.  Split the sentence

Line 74: modify the sentence (rewrite)

Line 42: Introduction is too long, please try to reduce it

Line 116: format mistake removes with subscript tool

Line 109: concise this portion

Line 137. ml or mL. please use the similar pattern.

Line 140: (as a reference) correct this sentence

Line 145: make a sense of this sentence

Line 148: change sentence, grammatically

Line 147: briefly describe this portion in a sequence, by choosing appropriate words

Line 166: minor changes or modification?

Line 169: subscript format

Line 243: mention treatment name either T1 or T2 compared with control treatment.

Line 253: use either abbreviations or both full texts instead of only one.

Line 379: if you have lot of studies to compare your results just chose and mention results of mango fruit. And mention treatment either which gives better results.

Line 386: discussion should be reduce if you compare results with only mango fruit instead of banana, orange, pine apple, because different fruits have different profile regarding to storage capacity with different treatment applications

Line 402: M&M needs to make it more concise

Line 512: old reference need to change, if possible

Line 546: too old reference, need to change with recent studies

Line 600: 82 references are too much, it should be maximum 40-45 for a good article.

Other comments:

You have used hot water as HW, please modify it throughout the manuscript. Please also check other abbreviations.

Please remove double spaces throughout the paper.

Please compare your results with those type of studies which have done storage studies. This comment is for all the results and discussion section.

Author Response

Reviewer # 4

  1. We modified the title by clearing combine treatment.
  2. Line 24: We concise the abstract.
  3. Line 29: We split the sentence.
  4. Line 78: We modified the sentence.
  5. Line 44: Introduction section, we reduced it.
  6. Line 141: We checked and corrected.
  7. Line 131: We concise material and method section.
  8. Line 163: We provided “ml” with the same pattern in all the manuscripts.
  9. Line 167-177: “as a reference” We checked and corrected.
  10. Line 171: We made sense for the sentence.
  11. Line 176-177: We checked and corrected the sentence, grammatically.
  12. Line 174: We briefly described the portion in a sequence, by appropriate words.
  13. Line 200: “with modification” not “minor changes”.
  14. Line 203: We checked and corrected the “subscript format”
  15. Line 349: We mentioned the treatment name and we removed abbreviation from all the manuscripts.
  16. Line 360: We used both full texts instead of only one, “Total soluble solids and titratable acidity”.
  17. Line 514-515: We chose and mentioned the results of mango fruit only. And mentioned treatment either which gives better results.
  18. Line 524: We reduced the discussion section and compared results with only mango fruit.
  19. I do not understand what is “M&M” in your comment but if it is material and methods, we reduced this section as you recommended.
  20. Line 190: It is difficult to omit this reference (Beers and Sizer) because Catalase (CAT) activity was measured using the method of this reference.
  21. Line 487: We omitted the old reference “Miller et al., 1991” and provided a recent study “Hasan et al, 2020).
  22. Line 698: We reduced the references number from 82 to 59. We provided a reference section with recent articles.

For other treatments

  1. We modified hot water treatment to HW throughout the manuscript. We also checked other abbreviations.
  2. We removed double spaces throughout the paper.
  3. In the result and discussion section: We provided new articles which have done storage studies and compared the results with those studies.

Round 2

Reviewer 1 Report

all data should be consisted, for instance, in table 3, "32.58, 13.8, 2.90", 12.8 should be 13.80?

Author Response

Reviewer #1#

The point: all data should be consisted, for instance, in table 3, "32.58, 13.8, 2.90", 12.8 should be 13.80?

Thanks for the value comments again. We have checked and corrected. (Please see Table 3)

Reviewer 2 Report

After the authors' revisions, the manuscript is worth of publication in Coatings

Author Response

The whole manuscript pars were revised and checked 

Reviewer 3 Report

Ref. No.: COATINGS-1520382-v2

Title: The use of hot water treatment and chitosan coating on mango (Mangifera Indica L. cv. Kent) fruits to control postharvest deterioration and increase fruit quality

Hoda A. Khalil, Diaa O. El-Ansary Lo’ay, A. A, Fatma K. M. Shaaban, Samah O. Osman, Ibrahim E. Shenawy, Hosam-Eldin Hussein Osman, Mohamed A. Abdein and Zinab A. Abdelgawad

Attempt to prolong and ensure quality and microbial safety of mango (Mangifera Indica L. cv. Kent) fruits, to control postharvest deterioration and increase fruit quality, authors applied hot water treatment and chitosan coating. Results indicate that combined treatment gave the best results and possess potential to be applied for mango fruit protection.

Still, there are plenty of papers indicated that both hot water and chitosan, in synergy, could be applied for different fruit protection. Idea and methodology of all those papers are similar. So, I don’t find novelty of this paper.

I found the paper to be overall well written and well described.

Title reflects the content of paper.

Authors used adequate literature for Introduction.

The experiment is clearly set up, and the analysis methods, their order, as well as description are mainly understandable.

The part Results was improved.

Major comments:

Technical mistakes:

  • Hoda A. Khalil, Diaa O. El-Ansary Lo’ay, A. A, Fatma K. M. Shaaban, Samah O. Osman; Ibrahim E. Shenawy, Hosam-Eldin Hussein Osman, Mohamed A. Abdein and Zinab A. Abdelgawad
  • Some references still have technical mistakes

I find authors accepted almost all my suggestions about technical mistakes and gave extra explanation I asked for.

The novelty of paper was not improved.

Author Response

#reviewer 3#

The point: Some references still have technical mistakes

We can feel the referee spent so much time reading our manuscript very carefully. We have checked and corrected all the technical mistakes in the reference section and provided DOI for all the references. We are sorry for the mistakes. Hopefully, the revised work is a much appropriate description. Thanks for the suggestions and we appreciate you giving us a great comment. (Please see the references section in the revised manuscript).

Reviewer 4 Report

Well revised. 

Author Response

Authors thank you very much for your efforts 

This manuscript is a resubmission of an earlier submission. The following is a list of the peer review reports and author responses from that submission.